# Angiogenesis in Chronic Inflammatory Skin Disorders

**DOI:** 10.3390/ijms222112035

**Published:** 2021-11-07

**Authors:** Hyun Ji Lee, Yong Jun Hong, Miri Kim

**Affiliations:** Department of Dermatology, Yeouido St. Mary’s Hospital, College of Medicine, The Catholic University of Korea, #10, 63-ro, Yeongdeungpo-gu, Seoul 07345, Korea; o0or5r5r@gmail.com (H.J.L.); yongjun2yo@hanmail.net (Y.J.H.)

**Keywords:** angiogenesis, psoriasis, atopic dermatitis, rosacea, chronic urticaria, hidradenitis suppurativa

## Abstract

Angiogenesis, the growth of new blood vessels from preexisting vessels, is associated with inflammation in various pathological conditions. Well-known angiogenetic factors include vascular endothelial growth factor (VEGF), angiopoietins, platelet-derived growth factor, transforming growth factor-β, and basic fibroblast growth factor. Yes-associated protein 1 (YAP) and transcriptional co-activator with PDZ-binding motif (TAZ) have recently been added to an important angiogenic factor. Accumulating evidence indicates associations between angiogenesis and chronic inflammatory skin diseases. Angiogenesis is deeply involved in the pathogenesis of psoriasis. VEGF, angiopoietins, tumor necrosis factor-a, interleukin-8, and interleukin-17 are unregulated in psoriasis and induce angiogenesis. Angiogenesis may be involved in the pathogenesis of atopic dermatitis, and in particular, mast cells are a major source of VEGF expression. Angiogenesis is an essential process in rosacea, which is induced by LL-37 from a signal cascade by microorganisms, VEGF, and MMP-3 from mast cells. In addition, angiogenesis by increased VEGF has been reported in chronic urticaria and hidradenitis suppurativa. The finding that VEGF is expressed in inflammatory skin lesions indicates that inhibition of angiogenesis is a useful strategy for treatment of chronic, inflammatory skin disorders.

## 1. Introduction

Angiogenesis is defined by the formation of new blood vessels from preexisting ones. This is essential both in utero and after birth. In adults, physiological angiogenesis is in play during the menstrual cycle and wound-healing [1]. Angiogenesis must be well-controlled both during development and in adulthood. Dysfunctional angiogenesis contributes to a variety of pathological conditions, including various infectious diseases, immune system and inflammatory disorders, and genetic diseases such as von Hippel-Lindau disease, cancer, retinopathy, and arteriosclerosis [2].

Accumulating evidence indicates associations between angiogenesis and inflammation in various pathological conditions. These two phenomena have long been observed to be coupled, exacerbating many different, chronic inflammatory diseases. The interplay between inflammatory and endothelial cells, and fibroblasts, in chronic inflammatory lesions, and the fact that inflammation and angiogenesis can be triggered by the same molecular events, further strengthens the evidence of the relationship. Elucidation of the cellular and molecular mechanisms that link the two processes is essential to understand their synergistic effects and for the development of novel therapeutic approaches. Herein, we summarize current understanding of the molecular mechanisms underlying angiogenesis in patients with chronic, inflammatory skin disorders, particularly psoriasis, atopic dermatitis, rosacea, urticaria, and hidradenitis suppurativa.

## 2. Molecular Mechanisms of Angiogenesis

Angiogenesis is initiated by specific growth factors. Folkman et al. found that tumor growth required de novo vascularization induced by a specific growth factor released from tumor cells during hypoxia [3]. This was subsequently termed vascular endothelial growth factor (VEGF) [4,5,6]. VEGF induces endothelial cell mitogenesis and migration, and promotes cell sprouting and vessel tube formation. The VEGF family includes VEGF-A, VEGF-B, VEGF-C, VEGF-D, and placental growth factor [6], but VEGF-A is thought to be the key regulator of angiogenesis during both homeostasis and disease. VEGF-A increases matrix-metalloproteinase (MMP) secretion and endothelial cell proliferation [7]. VEGF-C and VEGF-D are key lymphangiogenetic factors during development [8,9]. VEGF-E, which is structurally virtually identical to VEGF-A [10], and VEGF-F, were identified from snake venom [11]. An additional family member, VEGF-B, does not appear to exhibit angiogenic activity but is a key regulator of fatty acid metabolism [12].

The angiopoietin (Ang) family of stimulatory angiogenic factors includes Ang-1, Ang-2, Ang-3, and Ang-4. These molecules bind to an TEK endothelial receptor tyrosine kinase, Tie-2, to promote angiogenesis. Angs control endothelial cell homeostasis by modulating vascular maturation and stability, and cell survival [13]. VEGFs are activated in the early stage of angiogenesis whereas the Ang/Tie-2 systems are activated in later stages and control vessel assembly and maturation of the embryonic vascular system, as well as vessel homeostasis of the adult vascular system [14]. Ang-1 and Ang-2 act in an opposite manner. Ang-1 is a Tie2 receptor activator that maintains blood vessel formation by inducing endothelial-specific receptor Tie2 signaling. Ang-2 serves as an antagonist of Ang-1, destabilizing vessels by blocking Tie2 signaling. Ang-2 acts with VEGF to initiate angiogenesis. In addition, platelet-derived growth factor (PDGF) and transforming growth factor (TGF)-β (other angiogenic factors) stabilize new vessels. PDGF is important for maintenance of angiogenesis via recruitment of mural cells (principally pericytes). In addition, PDGF directly affects endothelial cells by inhibiting the angiogenic response to basic fibroblast growth factor (bFGF) [15]. TGF-β is responsible for production of the extracellular matrix (ECM) (a complex network of proteins, glycoproteins, polysaccharides, and proteoglycans). The ECM is directly or indirectly involved in angiogenesis, interacting with several growth factors and cytokines, and storing such factors. In addition, TGF-β upregulates VEGF, enabling sustained angiogenesis by stimulating endothelial cell proliferation, differentiation, and migration [16].

VEGF families induce endothelial regeneration and increase vascular permeability by binding to transmembrane receptor tyrosine-protein kinases (RTKs) termed vascular endothelial growth factor receptor (VEGFR)-1, -2, and -3. The VEGFR-1 ligands are VEGF-A and -B, and placental growth factor (PlGF). VEGFR-2 (known as the kinase insert domain receptor [KDR] in humans and fetal liver kinase 1 [Flk-1] in mice) ligands include VEGF-A, -C, and -D; VEGFR-2 is predominantly expressed in vascular endothelial cells. VEGFR2 exhibits the strongest affinity for VEGF-A and serves as the principal receptor for angiogenesis signaling [17]. In addition to the VEGFR-dependent pathway, several RTKs involved in angiogenesis are also known, including the fibroblast growth factor receptor (FGFR), ephrin, and PDGFR. Under low-oxygen conditions, these signaling pathways induce heterodimerization of the hypoxia-inducible factor-1 (HIF-1) transcriptional activator associated with adaptation to both cellular and organismal hypoxia. In addition, these pathways are involved in MMP activation, followed by ECM degradation. This is associated with release of other growth factors that stimulate endothelial cell migration.

The Hippo signaling pathway, a recent addition to the family of signaling pathways, is an evolutionarily conserved serine/threonine kinase signaling pathway that regulates tissue homeostasis and organ size by controlling cell proliferation, cell death/apoptosis, stem-cell self-renewal, and mechanotransduction [18,19,20,21]. The Hippo pathway also regulates (via phosphorylation) the activities of the downstream transcriptional co-activators Yes-associated protein 1 (YAP) and transcriptional co-activator with PDZ-binding motif (TAZ) [19]. On receipt of a wide range of signals induced by cell contact, polarity, energy metabolism, mechanical stress, and G-protein coupled receptor (GPCR) signaling, Hippo signaling is activated. Next, the mammalian STE20-like kinases (MST)1/MST2 and the large tumor suppressor kinase (LATS)1/LATS2 kinases are phosphorylated, in turn phosphorylating YAP/TAZ. This recruits the 14-3-3 proteins that promote cytoplasmic retention or proteolytic degradation. When Hippo signaling is inactive, YAP/TAZ become localized to the nucleus, where they form complexes with transcription factors of the TEA domain family to regulate genes required for endothelial cell proliferation, migration, and survival [22].

Endothelial changes are key features of early angiogenesis. Choi et al. showed that YAP was an important regulator of angiogenesis in the mouse. YAP is initially inactivated by phosphorylation and then redistributed in a cell contact-dependent manner by VE-cadherin. YAP knockdown in mice was associated with a significant decrease in the total tubular network and the number of endothelial sprouts in the aortic ring [23]. During angiogenesis, the VEGF-VEGFR2 signaling axis is absolutely dependent on activation of YAP/TAZ [24]. In human umbilical vein endothelial cells (HUVECs) and during post-natal development of the mouse retina, Ang2 is the key YAP target gene in endothelial cells; YAP regulation of angiogenesis and vascular remodeling is mediated by Ang2 [25]. Several receptors regulate YAP/TAZ activity directly (via LATS) or indirectly, to control angiogenesis. VEGFR regulates YAP/TAZ via the Rho GTPase, mitogen-activated protein kinase (MAPK), and phosphoinositide 3-kinase (PI3K) pathways [26,27,28,29]. The TGF-β [30,31,32,33], Wnt [34,35], and CD44 [36,37] pathways regulate YAP, TAZ, and LATS activity via mechanisms that are not yet understood. The angiogenetic factors are described in Table 1.

## 3. Psoriasis

Psoriasis is largely an immune system-mediated disease with both genetic and environmental predisposing factors. The prevalence is 1–3%. The key pathophysiology is immune cell-triggered keratinocyte hyperproliferation [54]. Although psoriasis is largely T-cell-driven [55], the pathophysiology is greatly modulated by abnormalities of the papillary dermal vasculature. The Auspitz (“bloody dew” [from German]) is a visible, characteristic, vascular abnormality that is pathognomonically diagnostic of psoriasis [56]. It appears as pinpoint bleeding that occurs when the scale of psoriatic plaque has been removed, reflecting vascular dilation and elongation with increased blood vessel permeability and a tortuosity specific for psoriasis [57]. Importantly, these vascular changes precede epidermal hyperplasia of psoriatic lesions. Psoriasis improvement (on appropriate treatment) is accompanied by normalization of the vascular structure [58,59], suggesting that psoriasis-associated microvascular abnormalities play functionally important roles in terms of the primary psoriasis pathogenesis.

Angiogenesis of psoriatic lesions (“inflammatory angiogenesis”) is characterized by significant vasodilation, vessel elongation, and increased vascular permeability [60,61]. In healthy skin, the dermal vessels exhibit principally the arterial phenotype, whereas the vessels of psoriasis evidence venous capillaries characterized by a single- or multi-layered basement membrane and a fenestrated endothelium that enhances vascular permeability [62]. Several angiogenic mediators such as VEGF, HIF-1α, the Angs, and the pro-angiogenic cytokines (including tumor necrosis factor [TNF], interleukin [IL]-8, and IL-17) are upregulated during psoriasis development [63]. Most cytokines are directly secreted by Th17, Th1, mast cells, macrophages, and neutrophils. Cytokine production is either directly induced as the psoriatic pathophysiology develops, or cytokine gene transcription is indirectly upregulated. VEGF and its receptors [38,39], CXCL8/IL-8 and TNFα [60], are upregulated in the keratinocytes of psoriatic lesions. TNF-α produced by mast cells, macrophages, keratinocytes, and lymphocytes seems to upregulate IL-8, VEGF, bFGF, Ang, and Tie-2 receptor expression in endothelial cells [40]. IL-17 secreted by Th17 cells not only directly promotes angiogenesis but also upregulates other angiogenic factors including VEGF and IL-8 [41,42]. IL-9 is critically involved in the VEGF-A-associated angiogenesis induced by IL-17 [64]. Recently, epidermal growth factor-like repeats and the discoidin I-like domain 3 (EDIL3) were reported to be highly expressed in the dermal mesenchymal stem cells of psoriasis. Using both in vitro and in vivo approaches, it was found that EDIL-3 promoted endothelial cell adhesion, migration, and tube formation. Therefore, EDIL-3 may play a role in the angiogenesis of psoriasis [65].

The VEGF-A expression level is higher in the lesional skin of psoriatic patients than in non-lesional or healthy skin [28,29,30]. Plasma levels of VEGF-A are higher in patients with psoriasis than in healthy individuals, and they correlate with disease severity [31,32]. VEGF-A is produced principally by activated keratinocytes in the skin of patients with psoriasis [29,33,43]. Other VEGFs (not produced by keratinocytes) are synthesized by fibroblasts [29] and immune cells such as mast cells [34,35]. Fibulin-3 (Fib3) is highly expressed in the keratinocytes and endothelial cells of psoriasis, contributing to angiogenesis by overexpressing VEGF [44]. In a mouse model of psoriasis, conditional deletion of VEGFR1 or neuropilin 1 (a VEGFA co-receptor amplifying VEGFA signaling in epidermal cells) inhibits psoriasis triggered by VEGF-A overexpression [66]. Both the Ang-Tie2 system and VEGF-A are closely associated with the microvascular proliferation of psoriasis. Reductions in the levels of these materials improve psoriatic lesions, suggesting that they play key roles in plaque vascular proliferation [51,52].

Several case reports on psoriasis treatment using angiogenesis inhibitors have appeared. Several compounds have been used to treat advanced renal cell carcinoma, including the PDGF receptors -α and -β, c-Kit, fms-like tyrosine kinase (Flt)-3, colony-stimulating factor receptor 1, glial cell line-derived neurotrophic factor receptor, and sunitinib, a multikinase inhibitor that targets VEGFR -1, -2, and -3. This was the first angiogenesis inhibitor to significantly improve chronic, large psoriatic plaques in psoriasis patients [67]. A Japanese case report found that sunitinib induces rapid but transient psoriasis improvement in patients with metastatic renal cell carcinoma [68]. Topical sunitinib ointment alleviates the clinical symptoms and reduces Ki-67 expression in an imiquimod-induced mouse model of psoriasis by modulating the levels of the cell cycle proteins D1 and E1, and poly ADP-ribose polymerase [69]. In addition, bevacizumab (a monoclonal antibody against VEGF that effectively treats various cancers, diabetic retinopathy, and retinal macular degeneration) induces remission of both psoriatic arthritis [70] and psoriasis [71]. Sorafenib, another (oral) multi-kinase inhibitor active against BRAF, CRAF, VEGFR, and PDGFR, clears the chronic psoriatic lesions of a 78-year-old male with hepatocellular carcinoma [72]. The angiogenesis in psoriasis is summarized in Figure 1.

Anti-angiogenesis therapies have become less popular, because anti-IL17A, anti-IL12/IL23, and anti-IL23 treatments have recently proved to be more effective with fewer adverse events. However, both the well-established and newly developed psoriasis treatments seek to perturb the complex cytokine network of angiogenesis. Anti-TNF treatment (adalimumab) considerably reduces endothelial cell proliferation, the vascular network size, and vessel diameter in psoriatic patients [73]. Infliximab (another anti-TNF agent) reduces the levels of pro-angiogenic factors such as VEGF, Ang-2, and TNF-α in cutaneous biopsy specimens of psoriatic patients [74]. Narrowband UVB irradiation, another treatment for psoriasis, reduces the serum levels of IL-8 and VEGF [75].

## 4. Atopic Dermatitis

Atopic dermatitis (AD) is a chronic, pruritic, inflammatory skin disease that is common in children and adolescents. Although the pathophysiology is not fully understood, many studies have demonstrated that AD is both complex and multifactorial, involving skin barrier dysfunction, cell-mediated immune response dysregulation, IgE-mediated hypersensitivity, and environmental factors. Defects in epidermal proteins that are maintained in the skin barrier allow allergens and microbes to penetrate into the skin. This is the first step in the “atopic march” of AD [76,77]. Immune dysregulation, including activation of type-2 immune responses, impair the epidermal barrier [78].

The histological features of acute AD include intercellular epidermis edema (“spongiosis”) and prominent perivascular dermal infiltration of lymphocytes, monocytes/macrophages, dendritic cells, and a few eosinophils. The subacute and chronic stages of AD are characterized by epidermal hyperkeratosis, acanthosis, and papillomatosis. At these stages, the dermal changes are less prominent than in the acute stage. All of these AD skin changes require angiogenesis [79]. VEGF levels in serum and skin are elevated in AD patients compared to controls; the rises correlate with AD severity as measured via the SCORAD instrument [80]. The VEGF levels are remarkably upregulated in the stratum corneum of lesional skin (compared to non-lesional skin) of AD patients [45]. In addition, an association between VEGF/VEGFR gene polymorphisms and AD has been reported [46].

The lesional skin inflammation of AD appears to be linked to vascular changes. Mast cells, basophils, eosinophils, macrophages, and lymphocytes are major sources of angiogenic and lymphangiogenic factors. Mast cells of AD lesions stimulate angiogenesis by releasing pro-angiogenic factors including VEGF-A and VEGF-B [46]. Interestingly, increased levels of the well-known angiogenic factors prostaglandin E2 and adenosine in AD induce VEGF-A and VEGF-B expression in human mast cells [47]. Such cells serve as targets for angiogenic factors; the cells express VEGFR-1 and VEGFR-2. VEGF secretion by mast cells is increased by the IL-9/IL-9 receptor pair, the level of which is elevated in AD [81].

Th17 cells play important roles in terms of clearing pathogens, and produce IL-17, IL-17F, IL-22, and IL-21 [82]. Th17 pathways are important in patients with chronic autoinflammatory diseases. IL-17+ cells are involved in psoriasis and have recently been reported to contribute to AD. A linear correlation is evident between disease severity and IL-17+ cell density [83]. Thus, IL-17+ cells aggravate AD by releasing angiogenic and proinflammatory factors.

Kim et al. showed that erythroid differentiation regulator-1 recombinant (rErdr1) administration improves AD. After such treatment, AD severity and the levels of immunoglobulin E and IL-4 significantly decrease in a mouse model [84]. In addition, the levels of C-C motif chemokine ligand (CCL)17 and CCL22 during AD angiogenesis are inhibited by rErdr1, reducing AD severity.

Recently, Cleo et al. reported that soluble VEGF receptor 1, a natural decoy inhibitor of VEGF-A, ameliorates skin lesions and inflammation in an AD model of APOC1 transgenic mice. Epidermal thickness and inflammatory infiltration are normalized, combined with decreases in the levels of IL-6 and the skin vascular cell adhesion molecule (VCAM)-1. Soluble VEGF receptor 1 could thus serve as a valuable AD treatment [85]. The angiogenesis in AD is summarized in Figure 2.

## 5. Rosacea

Rosacea is a chronic, inflammatory skin syndrome characterized by various combinations of signs and symptoms including erythema, telangiectasia, papules, and pustules on the central face; it affects 5–10% of the population. Although its pathophysiology is not fully understood, several factors have been implicated, including changes in the innate immune system, ECM degeneration, antimicrobial peptide dysfunction, and angiogenesis [86,87,88]. Certain specific receptors and channels are activated by bacterial proteases, demodex, heat, stress, irritants, and ultraviolet B radiation. All induce or exacerbate known rosacea-related factors, causing various rosacea phenotypes. The products of microbes including Demodex folliculorum and Staphylococcus epidermidis, and reactive oxygen species created via ultraviolet irradiation, are recognized by Toll-like receptor 2 (TLR 2). Subsequently, TLR 2 activates NACHT, LRR, and PYD domains-containing protein 3 (NALP3) inflammasome, which triggers kallikrein 5; such expression is also activated by MMPs [89]. Kallikrein 5 cleaves cathelicidin into LL-37, which then plays a role in the release of pro-inflammatory cytokines, chemokines, proteases, and pro-angiogenic factors; all mediate rosacea symptoms including erythema, telangiectasia, and inflammation. NALP3 inflammasome also activates mast cells, which produce inflammatory and angiogenetic factors, such as VEGF [90,91]. Other known triggers of rosacea, including spicy food, stress, exercise, and heat activate the transient receptor potential vanilloid receptor/transient receptor potential ankyrin 1 combination, which has been suggested to cause flushing and sensitivity. The angiogenesis in rosacea is summarized in Figure 3.

Hayran et al. recently showed that the VEGF gene was polymorphic [92]. In that study, the prevalence of the +405C/G polymorphism was higher in patients with erythematotelangiectatic, papulopustular, and phymatous rosacea than in controls, and the polymorphism was associated with rosacea severity. Lee et al. recently described a relationship between rosacea and the Hippo pathway that plays an important role in angiogenesis [48]. In immunohistochemical staining, YAP and TAZ were upregulated in rosacea patients. The characteristic features of rosacea were improved when a YAP/TAZ inhibitor was administered to mice with rosacea.

Doxycycline, minocycline, tetracycline, and brimonidine are extensively used in clinical settings to treat rosacea, given their anti-inflammatory and anti-vascular effects. Doxycycline is a broad-spectrum tetracycline-class antibiotic that inhibits the 30S ribosome subunit; the drug also has an anti-inflammatory effect [53]. Several reports on the effects of doxycycline on the ocular system [93], the oral cavity, intracranial venous hypertension [94], skin scarring [49], and rosacea have appeared. In patients with intracranial venous hypertension, doxycycline inhibits angiogenesis by reducing microvessel density, suppressing MMP-3 overexpression, and reducing VEGF and TGF-β levels [94]. In rosacea patients, doxycycline inhibits endothelial cell synthesis of MMP-8 and MMP-9, thus reducing cell migration during angiogenesis [95].

Brimonidine is a highly selective α2-adrenergic receptor agonist approved by the Food and Drug Administration for the topical treatment of rosacea [96]. It may directly vasoconstrict both small arteries and veins. Piwnica et al. [97] showed that brimonidine tartrate potently vasoconstricts vessels of diameter less than 200 µm in the human subcutis. The same group found that brimonidine tartrate inhibits edema in mouse models of ear inflammation. Kim et al. showed that rosacea induced by LL-37 in Balb/c mice improves after treatment with a topical brimonidine gel [98]. In that study, significant decreases in the number of mast cells and the levels of mRNA-encoding mast cell enzymes were apparent. A recent case report described successful treatment of rosacea after the application of broadband pulsed light and topical 0.5% (*w/v*) brimonidine tartrate; the rosacea had become progressively worse over the previous 8 years [99].

Topical dobesilate, an inhibitor of angiogenic growth factor, is an effective treatment for erythematotelangiectatic rosacea [100]. Erdr1, an anti-metastatic factor negatively regulated by IL-18, inhibits VEGF-mediated angiogenesis [87]. Artemisinin, an antimalarial drug from Artemisia annua L, exhibits anti-inflammatory and anti-angiogenic properties, and ameliorates rosacea-like dermatitis [101]. A recent study on mice found that aspirin reduces microvessel density and VEGF expression in rosacea-like skin, and also activation of NF-κB signaling and the release of downstream pro-inflammatory cytokines [102]. Tranexamic acid, an antifibrinolytic agent recently used to treat melasma in Asian patients, improves rosacea by reducing IL-6, TNFα, and MMP expression, and also lowers the angiogenesis of rosacea by reducing VEGF expression and the number of CD31+ cells [103].

In addition, interestingly, one study found that mild to severe papulopustular rosacea responds well to long-pulse neodymium:yttrium:aluminum:garnet laser treatment. In that study, follicular ablation and selective photothermolysis were apparent; these destroyed the telangiectasia and induced remodeling of dermal collagen [104]. In one study, radiofrequency irradiation improved rosacea induced by ultraviolet B in an animal model by reducing keratinocyte proliferation; it also improved the levels of pro-inflammatory cytokines, angiogenesis-related inflammatory factors, and VEGF, and attenuated the VEGF-induced pathophysiology of rosacea, reducing tube formation, cell migration, and endothelial cell proliferation [105].

## 6. Chronic Urticaria and Angioedema

Chronic urticaria (CU) is defined as the presence of pruritic and/or pricking wheals with erythema, angioedema, or both, for more than 6 weeks [106]. The prevalence ranges from 0.5% to 5%. CU is unpredictable in terms of both course and duration, persisting for several years in many patients. The wheals present with well-circumscribed non-pitting edema and blanched centers, and are usually surrounded by erythema; these are the characteristic signs of urticaria. Intracellular signaling defects and autoimmune processes activate mast cells and basophils, followed by spontaneous cellular degranulation associated with the release of principally histamine and other inflammatory mediators [107]. CU is commonly accompanied by angioedema of the deeper skin layers and subcutaneous tissue (resulting in diffuse swelling) or the submucosa of the upper respiratory and gastrointestinal tracts [106].

CU is associated with neo-vascularization and elevated vascular marker levels [108]. Lesional skin contains significantly more CD31-positive endothelial cells than normal skin. Confocal imaging has confirmed that urticarial lesions show increased vascularity. In addition, the increased numbers of new vessels and inflammatory cells are correlated. Tedeschi et al. found that the plasma levels of VEGF secreted by eosinophils, mast cells, and basophils are increased in patients with CU, and correlated with disease severity [109]. A recent study found that the sera of CSU patients induce mast cell production of VEGF via the PI3K/Akt/p38 MAPK/HIF-1α signaling pathway; 25 (OH)D3 suppresses VEGF expression by inhibiting signaling, suggesting that vitamin D treatment might control the angiogenesis of CU [50]. The angiogenesis in chronic urticaria is summarized in Figure 4.

Angioedema is a self-limiting but potentially life-threatening disorder characterized by edema in the deeper layers of the skin and mucosa attributable to periodic increases in vascular permeability caused by the release of bradykinin (BK) and/or other mast cell-derived mediators, including histamine. Hereditary angioedema is caused by a deficiency of, or a dysfunction in, the C1 esterase inhibitor (C1 INH) [110]. Angiogenesis plays an important role in its pathophysiology. VEGF-A and VEGF-C concentrations are increased in patients with hereditary angioedema; the levels correlate with disease severity [111]. Heparin secreted by mast cells induces expression of bradykinin, which in turn binds to and activates bradykinin receptor 2 of both mast and endothelial cells, increasing the levels of angiogenic/lymphangiogenic factors [112,113,114].

## 7. Hidradenitis Suppurativa

Hidradenitis suppurativa (HS) is a chronic inflammatory disease of skin, characterized by chronic and recurrent deep-seated nodules, abscesses, fistulae, and sinus tracts, eventually forming scars [115]. The most favorable sites are the axilla and inguinal area. Although the pathogenesis of HS has not been entirely elucidated, follicular hyperkeratosis within the pilosebaceous–apocrine unit is the first step of HS. Increased TNF-α from the keratocytes and activated dendritic cells and IL-17 from the Th17 cells are key cytokines in HS [116]. Furthermore, IL-1α was demonstrated to stimulate comedogenesis in the follicular infundibulum [117]. Because IL-1α is a potent inducer of the production of VEGF [118], it can be inferred that angiogenesis may play a role in the pathogenesis of HS. Derek J et al. recently reported that HS keratinocyte exhibited a significant lower level of VEGF, as well as IL-1α and IL-22 compared to normal keratinocyte using an in-vitro scratch assay, suggesting that changes in VEGF signaling may be associated with HS pathogenesis [119]. Furthermore, there was a case report that showed that sunitinib reactivates, worsens or triggers HS during treatment of a patient’s underlying cancer [120]. The association between HS and angiogenesis is still not well known, and further studies are needed in this area.

## 8. Conclusions

Angiogenesis is the process by which new blood vessels form from preexisting ones. Imbalanced angiogenesis contributes to many diseases. Herein, we focused on three chronic, inflammatory skin disorders: psoriasis, AD, and rosacea. Pro-angiogenic factors, VEGFs, and Ang-Tie system members, secreted by different immune cells, play key roles in blood vessel development and formation of the microvessel environment, by directly affecting various cell types. Recent evidence indicates that the YAP/TAZ system induces angiogenesis.

Psoriatic lesions feature markedly abnormal vascular networks including many enlarged, tortuous, and hyperpermeable cutaneous blood vessels. Secretion of various pro-angiogenic growth factors promotes vascular network expansion in psoriatic skin. In addition, pro-inflammatory cytokines activate endothelial cells and trigger pro-angiogenic actions. The angiogenesis of psoriasis is characterized by significant vasodilation, vessel elongation, increased vascular permeability, and inflammation of psoriatic lesional skin. AD is typically associated with a thickened epidermis (in the chronic phase) or intercellular edema (in the acute phase), and angiogenesis. As is true of psoriasis, pro-angiogenic growth factors secreted by activated immune cells induce angiogenesis. Rosacea, which is also characterized by neoangiogenesis, is induced by various pro-angiogenic factors and cytokines. Oral medications and laser therapy that regulate angiogenesis are now being implemented, as are other treatments. CU is also associated with increased vascularity and elevated vascular marker levels from eosinophils, mast cells, and basophils are increased in patients with CU and correlated with disease severity. In addition, it has been postulated that VEGF, along with IL-1α and IL-22, is associated with HS pathogenesis.

A better understanding of the molecular mechanisms in play, and the interactions between angiogenic factors and the endothelial cell environment, will foster the development of new therapeutic strategies for chronic inflammatory skin disorders.

## Figures and Tables

**Figure 1 ijms-22-12035-f001:**
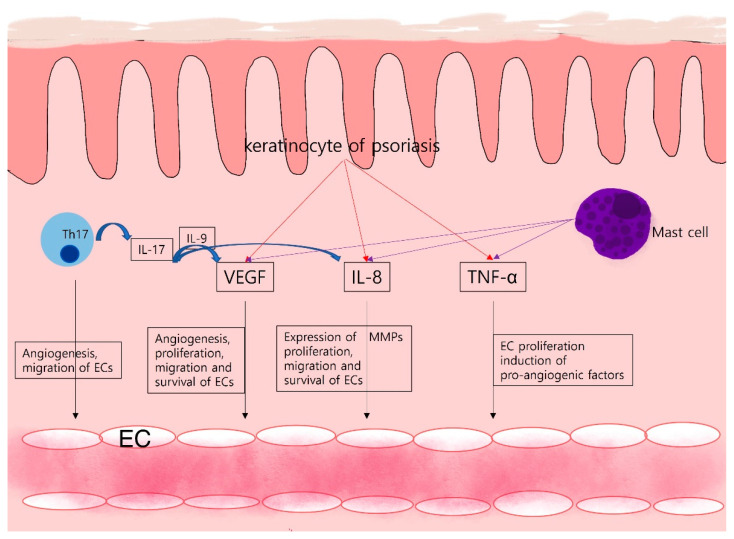
Angiogenesis in psoriasis: Th17 cells of psoriasis provoke angiogenesis through IL-17 production. In addition to angiogenesis promotion, expression of other angiogenic factors like VEGF and IL-8 is upregulated by IL-17. Keratinocytes, immune cells like mast cells, secrete a variety of pro-angiogenic factors and cytokines that activate and maintain the inflammatory skin condition of psoriasis.

**Figure 2 ijms-22-12035-f002:**
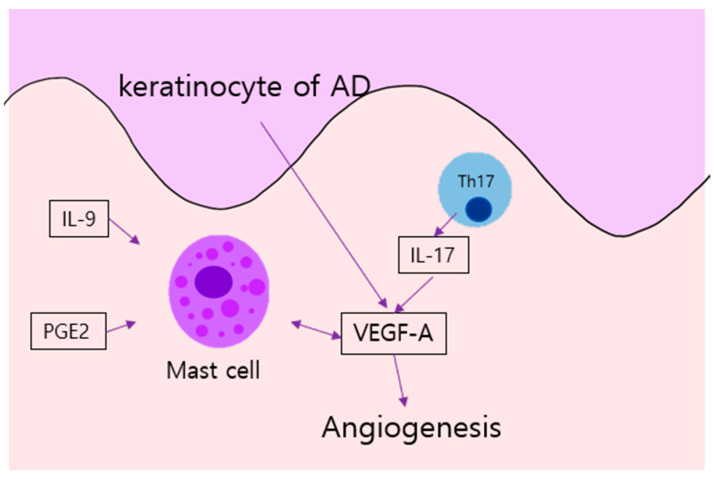
Angiogenesis in AD: Keratinocyte of AD produce VEGF. In addition, mast cells of AD, stimulated by IL-9 and PGE2, provoke angiogenesis through VEGF-A; such expression is also simulated by IL-17 from Th 17 cells. AD, atopic dermatitis; IL, interleukin, PGE2, prostaglandin E2; Th, T helper cell; VEGF, vascular endothelial growth factor.

**Figure 3 ijms-22-12035-f003:**
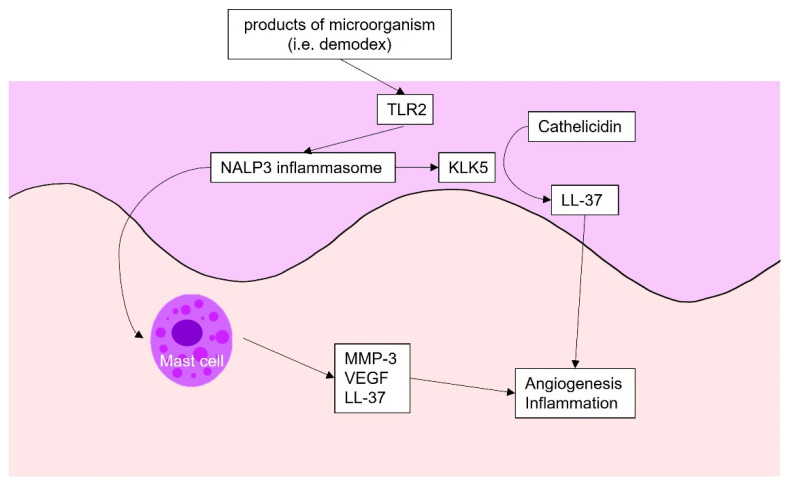
Angiogenesis in rosacea: The products of microbes are recognized by TLR 2. Subsequently, TLR 2 activates NALP3 inflammasome, which triggers kallikrein 5. Kallikrein 5 cleaves cathelicidin into LL-37, which triggers angiogenesis and inflammation. NALP3 inflammasome also activates mast cells, which produce inflammatory and angiogenetic factors, such as VEGF. TLR2, Toll-like receptor 2; KLK5, kallikrein5; MMP, metalloproteinase; VEGF, vascular endothelial growth factor.

**Figure 4 ijms-22-12035-f004:**
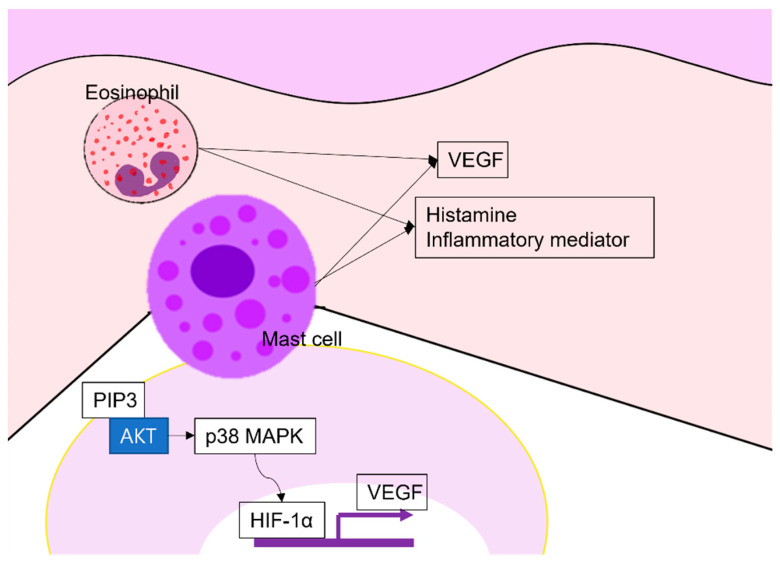
Angiogenesis in chronic urticaria: Eosinophils and mast cells produce VEGF, histamine, and other inflammatory mediators. Especially, mast cells produce VEGF via the PI3K/Akt/p38 MAPK/HIF-1α signaling pathway. VEGF, vascular endothelial growth factor; PIP3, phosphatidyl inositol 3,4,5 tri-phosphate; HIF, hypoxia-inducible factor-1.

**Table 1 ijms-22-12035-t001:** Angiogenetic factors, their receptor and functions, and related diseases. VEGF, vascular endothelial growth factor; VEGFR, vascular endothelial growth factor receptor; MMP, matrix-metalloproteinase; AD, atopic dermatitis; CU, chronic urticaria; Ang, angiopoietin; Tie, TEK receptor tyrosine kinase; PDGF, platelet-derived growth factor; PDGFR, platelet-derived growth factor receptor; TGF-β, transforming growth factor-β; EphR, Erythroprotein-producing human hepatocellular carcinoma receptors; YAP, Yes-associated protein 1; TAZ, Transcriptional co-activator with PDZ-binding motif.

Angiogenetic Factors	Receptors	Functions	Factor-Related Diseases
VEGF family	VEGFR-2 (main) [17] VEGFR-1	Induction of angiogenesis [4,5] Enhancement of vascular permeability and endothelial cell proliferation [7] Induction of MMP secretion [7]	Psoriasis [23,29,30,31,32,33,38,39,40,41,42,43,44] AD [45,46,47] Rosacea [48,49] CU [50]
Ang-1	Tie-2	Maintaining blood vessel formation [13] Stabilization of endothelial cell structure [14]	Psoriasis [40,51,52]
PDGF	PDGFR	Maintenance of angiogenesis, through recruiting mural cells, mainly pericytes [15]	
TGF-β		Production of extracellular matrix [16] up-regulates VEGF [16]	Rosacea [49]
bFGF	FGFR	Induction of angiogenesis [15]	Psoriasis [41]
YAP/TAZ	14-3-3 protein of Hippo pathway	Induction of VEGF/VEFR and Ang/Tie signaling pathway [23,24,25,26]	Rosacea [53]

## Data Availability

Not applicable.

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
