# Peer review of "Angiogenesis in Chronic Inflammatory Skin Disorders"

_ijms, 2021, doi:10.3390/ijms222112035_

Round 1
Reviewer 1 Report
This review is of potential interest to the readership of the Journal. However, an extensive editing of English language and style is necessary.
Additional comments:
Ln 62: "..promotes stabilization by formatting of endothelial tight junctions". Stabilization of what? Please specify. Also, what is "formatting" of tight junctions? Please explain. No information on that concept is available in the Ref. 14 cited to corroborate this statement.
Lns 66-67: “tumor growth factor (TGF)-beta” - should be “transforming growth factor (TGF)-beta”.
Ln 92: consider replacing “pre-existing” with “well-established”
Ln 197: consider replacing “of patients of psoriasis” with “of psoriasis patients”
Ln 239-240: The sentence “Although the importance…” is awkward and needs editing.
Ln 284: consider replacing “profusely” with “extensively”
Ln 287: should be “anti-inflammatory”
Lns 290-291: Rather than “overexpressing suppressed MMP-3”, should be “suppressing overexpressed MMP-3” (based on the data reported in the cited Ref. 92).
Lns 301-303: The sentence “Marked decrease…” is confusing and should be revised for clarity.
Lns 308-309: The authors state here that “Erdr1 … was observed to inhibit VEGF-mediated angiogenesis (83) and indirectly stimulate angiogenesis”. This is confusing. No reference is provided to support the latter statement. Please clarify.
Lns 375-377: The statement “Angiogenesis in psoriasis has a close association with… angiogenesis…” is redundant. Please revise.
Ln 379: “secretion of… growth factors and activated immune cells induce angiogenesis” – this sounds awkward, please rephrase (perhaps use “by” instead of “and”?)
Figure 1: needs to be improved. Some arrows seem to have been drawn by hand, contributing to a somewhat sloppy appearance of the figure.
Author Response
To Ziva Wang
Section Managing Editor
IJMS (International Journal of Molecular Sciences)
Re: [IJMS] Manuscript ID: ijms-1413728 entitled “Angiogenesis in chronic inflammatory skin disorders”
Thank you very much for your kind editorial letter.
We have attempted to carefully and thoroughly address all concerns raised by the editors and referees. With the help of your suggestions, we believe our manuscript has significantly improved.
The major changes are indicated by red font.
We trust that we therewith have fulfilled all the Editor’s and Reviewer’s requests.
Thank you very much for your consideration.
Sincerely,
Miri Kim, MD. PhD
Department of Dermatology, Yeouido St. Mary’s Hospital, College of Medicine, The Catholic University of Korea, #10, 63-ro, Yeongdeungpo-gu, Seoul 07345, Korea
E-mail: gimmil@naver.com
Reviewer 1
This review is of potential interest to the readership of the Journal. However, an extensive editing of English language and style is necessary.
Additional English editing service were provided. “The English in this document has been checked by at least two professional editors, both native speakers of English.”
For a certificate, please see:
http://www.textcheck.com/certificate/oeemBC
Additional comments:
Ln 62: "..promotes stabilization by formatting of endothelial tight junctions". Stabilization of what? Please specify. Also, what is "formatting" of tight junctions? Please explain. No information on that concept is available in the Ref. 14 cited to corroborate this statement.
Thank you for kind comment. I revised them in line 62-63.
“VEGFs are activated in the early stage of angiogenesis whereas the Ang/Tie-2 systems are activated in later stages and control vessel assembly and maturation of the embryonic vascular system, as well as vessel homeostasis of the adult vascular system.”
Lns 66-67: “tumor growth factor (TGF)-beta” - should be “transforming growth factor (TGF)-beta”.
Thank you for the comment. I revised them in line 67.
Ln 192: consider replacing “pre-existing” with “well-established”
Thank you for the comment. I revised them in line 200.
Ln 197: consider replacing “of patients of psoriasis” with “of psoriasis patients”
Thank you for the comment. I revised them in line 205.
Ln 239-240: The sentence “Although the importance…” is awkward and needs editing.
Thank you for the comment. I revised them in line 238-239.
Ln 284: consider replacing “profusely” with “extensively”
Thank you for the comment. I revised them in line 295.
Ln 287: should be “anti-inflammatory”
Thank you for the comment. I revised them in line 298.
Lns 290-291: Rather than “overexpressing suppressed MMP-3”, should be “suppressing overexpressed MMP-3” (based on the data reported in the cited Ref. 92).
Thank you for the comment. I revised them in line302.
Lns 301-303: The sentence “Marked decrease…” is confusing and should be revised for clarity.
Thank you for the comment. I revised them in line 310-311.
Lns 308-309: The authors state here that “Erdr1 … was observed to inhibit VEGF-mediated angiogenesis (83) and indirectly stimulate angiogenesis”. This is confusing. No reference is provided to support the latter statement. Please clarify.
Thank you for the comment. I revised them in line 317-318.
Lns 375-377: The statement “Angiogenesis in psoriasis has a close association with… angiogenesis…” is redundant. Please revise.
Thank you for the comment. I revised them in line 402-403.
Ln 379: “secretion of… growth factors and activated immune cells induce angiogenesis” – this sounds awkward, please rephrase (perhaps use “by” instead of “and”?)
Thank you for the comment. I revised them in line 405-406.
Figure 1: needs to be improved. Some arrows seem to have been drawn by hand, contributing to a somewhat sloppy appearance of the figure.
Thank you for king comment. I revised them in Figure 1.
Thank you again for your helpful review of our article.
With best regards,
Miri Kim, MD, PhD
Reviewer 2 Report
This is an interesting review regarding angiogenesis in skin diseases. I think you should include more diseases, at least you should talk about hidradenitis suppurativa
You could include some brief information regarding angiogenesis and the three disease in the abstract instead of talking about what is angiogenesis.
“With the finding that” Please change
“Angiogenesis defined as” Change to Angiogenesis IS defined by
“especially psoriasis, atopic dermatitis, rosacea, and urticaria”. In the abstract you only mention three of them. You should mention all.
Did you use any search criteria to conduct this review?
If you mention the The Auspitz phenomenon you should describe its clinical findings.
How did you selected the diseases included? It would be interesting to include information about hidradenitis suppurativa as it is another inflammatory skin condition and the potential role of VEGF-inhibitor have been also described
Doi: 10.1111/dth.13306
Doi: 10.1080/08820139.2017.1377227
I think it would be interesting to include a table that summarize the most important research in each disease
In the conclusion you only mention again three diseases. I think the conclusion could be summarized (line 371-383 can be omitted).
Author Response
To Ziva Wang
Section Managing Editor
IJMS (International Journal of Molecular Sciences)
Re: [IJMS] Manuscript ID: ijms-1413728 entitled “Angiogenesis in chronic inflammatory skin disorders”
Thank you very much for your kind editorial letter.
We have attempted to carefully and thoroughly address all concerns raised by the editors and referees. With the help of your suggestions, we believe our manuscript has significantly improved.
The major changes are indicated by red font.
We trust that we therewith have fulfilled all the Editor’s and Reviewer’s requests.
Thank you very much for your consideration.
Sincerely,
Miri Kim, MD. PhD
Department of Dermatology, Yeouido St. Mary’s Hospital, College of Medicine, The Catholic University of Korea, #10, 63-ro, Yeongdeungpo-gu, Seoul 07345, Korea
E-mail: gimmil@naver.com
Reviewer 2
This is an interesting review regarding angiogenesis in skin diseases. I think you should include more diseases, at least you should talk about hidradenitis suppurativa
Thank you for kind comment. I added hidradenitis supprativa in section 7.
You could include some brief information regarding angiogenesis and the three disease in the abstract instead of talking about what is angiogenesis.
“With the finding that” Please change
“Angiogenesis defined as” Change to Angiogenesis IS defined by
“especially psoriasis, atopic dermatitis, rosacea, and urticaria”. In the abstract you only mention three of them. You should mention all.
Thank you for kind comment. I revised them and added more information about diseases at abstract.
Did you use any search criteria to conduct this review?
The search terms were “angiogenesis and each dermatitis” and “vascular endothelial growth factor and each dermatitis' in pubmed. We has focused on research in the last five years.
If you mention the The Auspitz phenomenon you should describe its clinical findings.
Thank you for kind comment. I added description in line 133-136.
“It appears as pinpoint bleeding that occur when the scale of psoriatic plaque has been re-moved, reflecting vascular dilation and elongation with increased blood vessel permeabil-ity and a tortuosity specific for psoriasis.”
How did you selected the diseases included? It would be interesting to include information about hidradenitis suppurativa as it is another inflammatory skin condition and the potential role of VEGF-inhibitor have been also described
Doi: 10.1111/dth.13306
Doi: 10.1080/08820139.2017.1377227
Thank you for the comment. I added hidradenitis supprativa in section 7.
I think it would be interesting to include a table that summarize the most important research in each disease
Thank you for the comment. I added Table 1 that summarizing angiogenic factors and their related diseases.
In the conclusion you only mention again three diseases. I think the conclusion could be summarized (line 371-383 can be omitted).
Thank you for kind comment. I added conclusion in line 409-412.
Thank you again for your helpful review of our article.
With best regards,
Miri Kim, MD, PhD
Reviewer 3 Report
The authors should present the schemes individually, showing how psoriasis, atopic dermatitis, rosasea, and urticaria is regulated by the angiopoietic factors in each disease .
The authors should show the table summarizing the individual angiogenetic factors, their receptors, their functions, their target cells, the factor-related diseases, and references.
Author Response
To Ziva Wang
Section Managing Editor
IJMS (International Journal of Molecular Sciences)
Re: [IJMS] Manuscript ID: ijms-1413728 entitled “Angiogenesis in chronic inflammatory skin disorders”
Thank you very much for your kind editorial letter.
We have attempted to carefully and thoroughly address all concerns raised by the editors and referees. With the help of your suggestions, we believe our manuscript has significantly improved.
The major changes are indicated by red font.
We trust that we therewith have fulfilled all the Editor’s and Reviewer’s requests.
Thank you very much for your consideration.
Sincerely,
Miri Kim, MD. PhD
Department of Dermatology, Yeouido St. Mary’s Hospital, College of Medicine, The Catholic University of Korea, #10, 63-ro, Yeongdeungpo-gu, Seoul 07345, Korea
E-mail: gimmil@naver.com
Reviewer 3
The authors should present the schemes individually, showing how psoriasis, atopic dermatitis, rosasea, and urticaria is regulated by the angiopoietic factors in each disease .
Thank you for the comment. We added them as figure 2, figure 3, and figure 4.
The authors should show the table summarizing the individual angiogenetic factors, their receptors, their functions, their target cells, the factor-related diseases, and references.
Thank you for the comment. We added Table 1 as you recommended.
Thank you again for your helpful review of our article.
With best regards,
Miri Kim, MD, PhD
Round 2
Reviewer 1 Report
The manuscript is much improved.
Minor issues:
Ln 17-18 currently "... of rosacea", should be "... in rosacea"
Ln 21: currently "...inhibition of angiogenesis usefully treats..". I would suggest to revise as follows: "..inhibition of angiogenesis is a useful strategy for treatment..."
Author Response
Thank you very much for your kind editorial letter.
The changes are indicated by red font.
We trust that we therewith have fulfilled all the Editor’s and Reviewer’s requests.
Thank you very much for your consideration.
Sincerely,
Miri Kim, MD. PhD
Department of Dermatology, Yeouido St. Mary’s Hospital, College of Medicine, The Catholic University of Korea, #10, 63-ro, Yeongdeungpo-gu, Seoul 07345, Korea
E-mail: gimmil@naver.com
Ln 17-18 currently "... of rosacea", should be "... in rosacea"
-> Thank you for the comment. I revised them in line 17-18.
Ln 21: currently "...inhibition of angiogenesis usefully treats..". I would suggest to revise as follows: "..inhibition of angiogenesis is a useful strategy for treatment..."
-> Thank you for the comment. I revised them in line 21.
Reviewer 3 Report
The authors well addressed the issues I pointed out and appropriately revised the manuscript.
Author Response
Thank you very much for your kind editorial letter.
Sincerely,
Miri Kim, MD. PhD
Department of Dermatology, Yeouido St. Mary’s Hospital, College of Medicine, The Catholic University of Korea, #10, 63-ro, Yeongdeungpo-gu, Seoul 07345, Korea
E-mail: gimmil@naver.com